

# Effects of foliar applied asparagine, glycine and citric acid on cadmium uptake and yield in wheat

Özlem Ete Aydemir

Department of Soil Science and Plant Nutrition, Faculty of Agriculture, Ordu University, Ordu, Turkey

## ABSTRACT

Cadmium (Cd) contamination is an important heavy metal that threatens agricultural production and food safety. This study investigates the potential of foliar applications of asparagine (Asn), glycine (Gly) and citric acid (CA) to reduce Cd uptake and improve the nutrient balance of wheat (*Triticum durum* L.). A pot experiment was conducted under controlled conditions with three different Cd concentrations (0, 3 and 12 mg Cd $kg^{-1}$ soil). The mixture containing 0.5 mMg Gly, Asn, and CA was applied to the leaves a total of five times at 5-day intervals during the tillering, stem elongation, heading and flowering periods of wheat, starting 39 days after planting. The analysis showed that the application of Asn reduced the Cd concentration in the grains by 14.82%, 31.08% and 16.66%, respectively, while the application of Gly resulted in a reduction of 37.78%, 16.41% and 12.79% and the application of CA resulted in a reduction of 34.78%, 36.25% and −1.60% compared to the control (C0) group. In addition, grain yield increased in response to the amino acid applications, with Asn improving yield by 6.10%, 9.95% and 5.90%; glycine by 3.86%, 7.59% and 9.34%; CA by −2.64%, 6.16% and 3.44%, respectively. These amino acid treatments alleviated the growth limitations caused by Cd stress by increasing the grain yield of wheat. However, the effect of CA on Cd detoxification was lower compared to the amino acids. The results show that Asn and Gly not only reduce Cd accumulation in wheat but also improve nutrient balance and increase yield. Consequently, foliar application of these amino acids is a promising strategy to improve plant safety in Cd-contaminated agricultural soils.

## INTRODUCTION

Agricultural land is a non-renewable natural resource (*Kocaman, 2023a*). Therefore, environmentally sustainable agricultural development is essential to achieve the United Nations Sustainable Development Goals (*Hou et al., 2020*). However, a significant proportion of the world's agricultural land is contaminated with Cadmium (Cd), and other heavy metals (*Rahman & Singh, 2019*). Among these metallic pollutants, Cd contamination has become a globally significant environmental problem due to industrial and agricultural activities. The Cd contamination in agricultural soils is a significant global environmental issue, with severe implications for human health and agricultural productivity. In China,

Corresponding author
Özlem Ete Aydemir,
ozlemete87@gmail.com

extensive assessments reveal that approximately 200,000 km$^2$ of soil—equivalent to about one-sixth of the total farmland are impacted by heavy metal pollution, with Cd being a prominent contaminant (*He et al., 2025*). Rice, a staple food for many populations, is known to bioaccumulate Cd, which can lead to severe health issues, including renal damage and increased risk of chronic diseases (*Zhang et al., 2018*; *Karpinska et al., 2024*). The situation is especially critical in areas like the North China Plain, where practices such as sewage irrigation have exacerbated contamination levels (*Song et al., 2021*; *Yeshiwas et al., 2025*). Cd is a non-essential heavy metal that enters the soil through various anthropogenic sources, including mining, metal smelting, industrial wastewater discharges, phosphate fertilizers, sewage sludge application, landfill leachate (*Koptsik, 2014*; *Gülmezoğlu, Kutlu & Sağır, 2023*; *Kocaman et al., 2024*,), and atmospheric deposition (*Alloway, 2012*). In particular, the widespread use of phosphate-based fertilizers contributes significantly to the accumulation of Cd in soils, as phosphate rock naturally contains varying amounts of Cd (*Lugon-Moulin et al., 2006*; *Korkmaz et al., 2017*). In addition, irrigation with contaminated water and the long-term use of pesticides and fungicides can further increase the Cd content in agricultural soils (*Gill & Tuteja, 2010*). Once Cd has accumulated in the soil, it remains permanently due to its low mobility and strong affinity to organic material and clay minerals (*McLaughlin et al., 2006*; *Özkutlu et al., 2007*). Unlike essential micronutrients, Cd does not participate in the biological cycle but accumulates over time, which increases the risk of uptake by plants (*Rizwan et al., 2016*). The availability of Cd in soil is influenced by factors such as soil pH (*Cheraghi, Lorestani & Merrikhpour, 2011*), organic matter content (*Lugon-Moulin et al., 2006*), cation exchange capacity (*Carne et al., 2021*), and competing ions like zinc (Zn) and calcium (Ca) (*Özkutlu & Kara, 2018*; *Chen et al., 2022*). Acidic soils increase the bioavailability of Cd, facilitate its uptake by plant roots, and increase the likelihood of Cd accumulation in edible plants (*Korkmaz et al., 2010*; *Li et al., 2024*).

Durum wheat (*Triticum durum* L.) is an important agricultural crop, particularly in areas prone to Cd contamination. Compared to other wheat species, durum wheat generally accumulates higher levels of Cd, leading to significant yield reductions and posing a considerable risk to food safety (*Vergine et al., 2017*; *Yan et al., 2020*; *Fotovat, Abedinzadeh & Atarodi, 2024*; *Özkutlu 2024*). Elevated Cd concentrations can inhibit the uptake of essential micronutrients such as Zn, iron (Fe), and manganese (Mn), thereby negatively affecting the nutritional quality of durum wheat grains (*Rengel & Graham, 1995*). Although durum wheat plants develop various physiological mechanisms to limit Cd translocation, these mechanisms often become inadequate as soil Cd concentrations increase, resulting in grain Cd levels exceeding the regulatory limits set for human consumption (*Vergine et al., 2017*; *Nicaise et al., 2022*). Notably, significant variability in Cd accumulation among durum wheat varieties suggests that genetic factors play a critical role in plant responses to Cd stress (*Greger & Löfstedt 2004*; *Carne et al., 2021*; *Sabella et al., 2021*). Additionally, soil physicochemical properties and agricultural practices may also influence Cd uptake by plants (*Calzarano et al., 2018*; *Woźniak, Nowak & Gawęda, 2021*). Thus, investigating the effectiveness of innovative agricultural practices, such as the application of amino acids and organic acids, appears promising for enhancing durum wheat yield and grain quality in Cd-contaminated soils.

The presence of Cd in wheat grains poses a major problem for food safety, both for human and animal health, as Cd is a toxic element with no biological function. Chronic exposure to contaminated food can lead to serious health problems, including kidney failure, bone mineral loss, cardiovascular disease, and carcinogenic effects (*Al-Taani et al., 2021*; *Guo, Li & Xu, 2023*). To protect public health, international regulatory bodies such as the Codex Alimentarius Commission, the European Union, and the United States Food and Drug Administration (FDA) have set strict maximum levels for Cd in food. The European Commission, for example, has set the maximum Cd concentration in wheat grains at 0.2 mg Cd kg$^{-1}$ (*Alloway, 2012*; *Molina et al., 2021*). Therefore, it is crucial to develop effective mitigation strategies to prevent Cd accumulation in wheat grains. Various agricultural practices have been proposed to limit the bioavailability of Cd, including soil amendments, crop rotation, and the selection of varieties with low Cd accumulation (*Fu et al., 2022*; *Mubeen et al., 2023*; *Li et al., 2024*; *Cahyaningrum, Priyandani & Gusti, 2024*). However, most of these strategies require long-term implementation and do not provide immediate solutions for farmers growing wheat on contaminated soils. Therefore, emerging studies suggest that foliar application of amino acids (AAs) and organic acids (OAs) can mitigate Cd stress by modulating metal uptake and promoting plant resilience. (*Singh et al., 2016*; *Li & Wertheimer, 2023*; *Cahyaningrum, Priyandani & Gusti, 2024*; *Scott & Wu, 2024*). However, the comparative effectiveness of different AAs and OAs remains poorly understood.

It is known that phytohormones play a crucial role in the growth and development of plants. Therefore, the exogenous application of plant hormones as a physico-chemical approach can have a positive effect on improving the growth activity of plants under heavy metal stress (*Kocaman, 2023b*), as they act as chemical messengers in the plant defense system (*Sytar et al., 2019*). AAs and OAs also play an important role in plant metabolism and stress tolerance. Exogenous administration of specific AAs such as asparagine (Asn), and glycine (Gly), which were used in this study, regulates metal uptake and activates detoxification processes. In addition, these compounds have been reported to increase the resistance of plants to heavy metal stress by strengthening antioxidant defense mechanisms (*Gill & Tuteja, 2010*; *Kocaman, 2023a*). Similarly, citric acid (CA), a natural chelating agent, has been reported to reduce the bioavailability and toxicity of Cd by affecting the mobility and sequestration of Cd in plant tissues (*Sun et al., 2005*; *Ehsan et al., 2014*; *Kocaman, 2023a*). Therefore, foliar application of AAs and OAs is a promising strategy to reduce Cd accumulation without altering soil properties. In contrast to soil amendments, foliar sprays deliver nutrients and protective compounds directly to the plant tissue, thus reducing Cd uptake *via* the roots (*Singh et al., 2016*; *Lei et al., 2024*; *Wang et al., 2024*). Research has shown that foliar applications based on AAs can support plant growth, improve nutrient uptake, and reduce the transfer of heavy metals to edible plant parts (*Bashir et al., 2018*; *Ali et al., 2022*; *Aydemir, Korkmaz & Özkutlu, 2022*).

This study aims to mitigate the deleterious effects of Cd by exogenous foliar application of AAs (Asn, Gly) and an OA (CA). In addition, the potential of these treatments as an innovative and practical strategy for the production of safe agricultural products will be investigated. Based on the study results, the effectiveness of these applications in terms of

**Table 1  Some physical and chemical properties of the soil used in the experiment.**

| Parameters | Unit | Value |
|---|---|---|
| Sand | % | 8.6 |
| Silt | % | 30.8 |
| Clay | % | 60.6 |
| Texture | | Clay |
| pH | 1:2.5 | 8.08 |
| Electrical conductivity (EC) | mmhos/cm | 0.22 |
| Total CaCO$_3$ | % | 14.2 |
| Organic matter | % | 0.7 |
| DTPA-Zn | mg kg$^{-1}$ | 0.1 |
| Total Zn | mg kg$^{-1}$ | 51 |
| DTPA-Cd | mg kg$^{-1}$ | 0.005 |
| Total Cd | mg kg$^{-1}$ | 0.27 |

Cd accumulation, grain yield, nitrogen (N) and microelement uptake will be evaluated to gain practical insights into how effectively these treatments can improve the safety and productivity of wheat under Cd stress. In this study, N was selected for analysis because the main objective was to investigate the relationship between Cd stress and N metabolism, particularly its effect on AA and protein synthesis (*Boussama et al., 1999*; *Zhu et al., 2018*). Previous studies have reported that Cd toxicity disrupts N uptake and metabolism in plants, which in turn negatively affects plant growth and grain quality (*Yang et al., 2020*). Therefore, N was chosen as a representative macronutrient to evaluate the physiological effects of foliar treatments under Cd stress.

## MATERIAL AND METHODS

The experiment was carried out in a greenhouse under natural daylight conditions. The geographical coordinates of the greenhouse are 37°03′21″N and 35°21′18″E, at an altitude of 127 m above sea level in Türkiye. The internal temperature of the greenhouse ranged from 15 °C to 25 °C, depending on seasonal and diurnal variations. A completely randomized design (CRD) was employed to ensure unbiased distribution of treatments, with three replications per treatment group. Each pot constituted an experimental unit, and treatments were randomly assigned across all pots to minimize positional effects. The soil used in the experiment was collected from the topsoil layer (0–30 cm) of a farmer-managed agricultural field, selected to represent typical regional cultivation conditions. After collecting, the soil was air-dried, homogenized, and passed through a 4 mm sieve prior to use. Selected physicochemical properties of the soil are presented in Table 1. The experiment was conducted using the durum wheat (*Triticum durum* L.) cultivar Harran-95, which is widely cultivated in the study region.

The durum wheat variety Harran-95 (*Triticum durum* L.) was selected for this experiment due to its wide cultivation in Southeastern Anatolia, Türkiye, and proven agronomic performance under abiotic stress conditions such as drought and salinity. It is a spring-type cultivar that does not require vernalization to flower and completes its
life cycle under short to moderate cold exposure, making it suitable for autumn sowing in Mediterranean and semi-arid climates (*Kilic & Yağbasanlar, 2010*). The variety is well-adapted to regional agroecological conditions and is commonly used in both research and production systems in the region.

The seeds were sown in number 8 plastic pots (35 cm diameter) filled with 2.65 kg of soil each. The following nutrients and Cd doses were added uniformly to the test soil (per kg of dry soil): 100 mg phosphorus (P) in the form of $KH_2PO_4$, 2.5 mg Fe in the form of ethylenediaminetetraacetic acid (Fe-EDTA), 200 mg N in the form of $Ca(NO_3)_2.4H_2O$ and 1 mg Zn in the form of $ZnSO_4.7H_2O$ per kg of soil were applied as basic fertilizer. Three different doses of Cd (0, 3, and 12 mg Cd kg$^{-1}$ soil) in the form of $(CdSO_4)_3.8H_2O$ were used as application doses. Ten seeds were sown in each pot. Shortly after emergence, the seedlings were thinned to four per pot, depending on the experiment. Throughout the experiment, the plants were irrigated with deionized water daily under greenhouse conditions. The irrigation regime was planned by considering the physical properties of the soil and the pot volume, ensuring that the soil moisture was maintained close to field capacity. This approach prevented the plants from being exposed to any water stress. The soil used in the experiment had a clay-loam texture, which is known for its high water-holding capacity. Therefore, the daily irrigation amount was carefully adjusted based on the amount of soil in each pot, and each pot received approximately 142 ± 5 mL of deionized water per day. This strategy helped to prevent over-irrigation and the leaching of nutrients. The plants were grown at three different Cd inputs into the soil until grain maturity and harvested at grain maturity (day 96).

## Foliar sprays with different amino acids

The control treatments were sprayed with distilled water and 0.02 Tween-20 (w/w). Tween-20 was used as a surfactant in the application solution to facilitate penetration into the leaves and uptake of the foliar-applied amino acids. A foliar application of 0.5 mM of various amino acids was first applied on the 39th day after sowing, then at 5-day intervals, and finally on the 67th day after sowing. 0.5 mM Gly, Asn, and CA (w/v) were applied five times at 5-day intervals during the period of tillering, stem growth, sprouting, and ear/flowering of the wheat. All foliar application solutions contained 200 mg L$^{-1}$ Tween20 as surfactant and the plants were sprayed with a hand sprayer until runoff. Plants were harvested at grain maturity (day 96) and dried at 60 °C to determine the dry weight of the grains in the ears. The grains were harvested manually, and the grain yield was determined.

## Element analysis

The dried grain samples were ground to a fine powder using a clean stainless-steel grinder and stored in polyethylene containers. Approximately 0.2 g of each sample was subjected to acid digestion using 5 mL of 65% (v/v) $HNO_3$ and 2 mL of 30% (v/v) $H_2O_2$ in a closed-vessel microwave digestion system (MarsExpress; CEM Corp., Matthews, NC, USA). The digestion process was conducted in Teflon vessels that were thoroughly rinsed with deionized water and acid-washed to prevent cross-contamination. All labware used in the analysis was soaked in 10% (v/v) nitric acid for at least 24 h and rinsed with ultrapure water prior to use.

**Table 2  The effects of foliar application of amino and organic acids on the physico-chemical properties of wheat plants.**

| Cd (mg kg$^{-1}$) | Treatments | Yield g plant$^{-1}$ | N% % | Cd μg kg$^{-1}$ | Fe | Zn | Cu | Mn |
|---|---|---|---|---|---|---|---|---|
| | | | | | mg kg$^{-1}$ | | | |
| Cd0 | C0 | 4.92 ± 0.54 | 1.74 ± 0.18b | 62 ± 9a | 37.6 ± 3.7b | 10.4 ± 1.4b | 4.2 ± 0.3a | 37.4 ± 3.3a |
| | Asn | 5.22 ± 0.46 | 2.23 ± 0.17a | 54 ± 4ab | 44.2 ± 4.6ab | 14.5 ± 1.7a | 2.6 ± 0.2b | 27.4 ± 2.8bc |
| | Gly | 5.11 ± 0.32 | 1.85 ± 0.11b | 45 ± 1.70b | 47.6 ± 3.2a | 13.7 ± 0.8a | 2.9 ± 0.25b | 31.4 ± 2.8b |
| | CA | 4.79 ± 0.55 | 1.78 ± 0.18b | 46 ± 2.2b | 45.9 ± 3.3a | 11.7 ± 1.8ab | 2.4 ± 0.2b | 24.9 ± 3.7c |
| Cd3 | C0 | 4.22 ± 0.48 | 1.84 ± 0.15b | 9,826 ± 22a | 39.3 ± 2.3b | 11.8 ± 1.1b | 5.1 ± 0.3a | 37.5 ± 1.1a |
| | Asn | 4.64 ± 0.55 | 2.11 ± 0.11a | 7,496 ± 25c | 55.7 ± 0.5a | 16.1 ± 1.3a | 4.5 ± 0.1b | 28.7 ± 1.1b |
| | Gly | 4.54 ± 0.36 | 1.94 ± 0.12ab | 8,441 ± 12b | 52.1 ± 3.2a | 13.1 ± 0.9b | 3.9 ± 0.2c | 29.1 ± 0.9b |
| | CA | 4.48 ± 0.48 | 1.99 ± 0.13ab | 7,212 ± 12c | 51.7 ± 4.9a | 12.9 ± 1.2b | 3.7 ± 0.2c | 26.1 ± 1.2c |
| Cd12 | C0 | 4.07 ± 0.42 | 1.78 ± 0.12 | 13,634 ± 19a | 46.4 ± 4.2b | 13.8 ± 1.5b | 4.5 ± 0.6a | 39.3 ± 3.1a |
| | Asn | 4.31 ± 0.32 | 2.01 ± 0.14 | 11,687 ± 28b | 58.7 ± 4.3a | 15.1 ± 1.1ab | 2.2 ± 0.2c | 28.3 ± 2.9b |
| | Gly | 4.45 ± 0.38 | 1.81 ± 0.11 | 12,089 ± 34b | 54.6 ± 4.8ab | 15.5 ± 1.5ab | 2.9 ± 0.31b | 29.4 ± 1.23b |
| | CA | 4.21 ± 0.29 | 1.89 ± 0.11 | 13,852 ± 23a | 54.3 ± 3.7ab | 16.5 ± 1.3a | 2.8 ± 0.1bc | 25.94b |

Notes.

According to Duncan's multiple range test, data followed by different letters are significantly different ($p < 0.05$). Values without letters indicate no significant difference ($p > 0.05$).

Following digestion, the solutions were filtered and brought to a final volume with ultrapure deionized water. The concentrations of Zn, Fe, copper (Cu), Mn, and Cd were determined using inductively coupled plasma optical emission spectrometry (ICP-OES) (Vista-Pro Axial, Varian Pty Ltd, Mulgrave, Australia). Instrument calibration was performed using multi-element standard solutions, and accuracy was validated using a certified reference material (wheat flour SRM 1567a, National Institute of Standards and Technology, Gaithersburg, MD, USA). Each batch of samples included procedural blanks and duplicates to ensure data quality and detect any potential contamination.

The total N% concentration in the grains was determined separately using the Kjeldahl method (*Bremner, 1965*).

## Statistical analysis

The data were evaluated using a two-way analysis of variance (two-way ANOVA) conducted with SPSS software (version 22.0; SPSS, Chicago, IL, USA). To identify differences between groups, Duncan's multiple range test was applied, with the significance level set at $p < 0.05$. Values marked with different letters indicate statistically significant differences, while values without letters represent groups with no significant differences.

## RESULTS

In this study, the effects of foliar applications of 0.5 mM aAsn, Gly, and CA on grain yield, N concentration, Cd accumulation, and mineral content (Fe, Zn, Cu, Mn) in wheat grains under different Cd doses (0, 3, and 12 mg Cd kg$^{-1}$) were analyzed (Table 2).

The grain yield varied depending on the Cd concentration in the culture medium and the amino acid application on the leaf; however, these differences were not statistically significant. In the control groups (C0), the highest grain yield was observed under Cd0

conditions (4.92 g plant$^{-1}$), while the lowest yield was recorded under Cd12 conditions (4.07 g plant$^{-1}$). Under Cd0 conditions, all amino acid applications increased grain yield compared to the control group. The highest yield increase was observed with the application of Asn, with yield increases of 6.10%, 9.95% and 5.90% for plants grown under Cd0, Cd3, and Cd12 soil conditions, respectively.

The N accumulation in the wheat grains showed statistically significant differences in response to the amino acid applications ($p < 0.05$). The lowest N content was found in the C0 groups, while the highest N content was obtained when Asn was applied. However, under Cd12 conditions, the N content in the wheat grains showed no statistically significant differences in response to the amino acid applications. Compared to the C0 groups, the highest N content was observed with the application of Asn, with increases of 28.16%, 14.67% and 12.92% under Cd0, Cd3, and Cd12 conditions, respectively.

As a result of increasing Cd doses and amino acid applications, it was determined that there were statistically significant differences between grain Cd concentrations ($p < 0.05$). In the Cd0 group, decreases occurred in grain Cd concentrations with amino acid applications compared to the control. While the grain Cd concentration in the control was $62 \pm 9$ µg kg$^{-1}$, the highest decrease was determined as $45 \pm 1.70$ µg kg$^{-1}$ in the Gly application. In the Cd3 dose, while the grain Cd concentration in the control was $9,826 \pm 22$ µg kg$^{-1}$, it was determined that there were decreases compared to the control due to AAs and OAs applications. Grain Cd concentrations were determined to be $7,496 \pm 25$ µg kg$^{-1}$ in Asn, $7,212 \pm 12$ µg kg$^{-1}$ in CA and $8,441 \pm 12$ µg kg$^{-1}$ in Gly. As the applied Cd amount increased, Cd accumulation in the grain increased. While grain Cd concentration was $13,634 \pm 19$ µg kg$^{-1}$ in the control at the Cd12 dose, the decreases in grain Cd concentrations in Asn and Gly treatments were $11,687 \pm 28$ µg kg$^{-1}$ and $12,089 \pm 34$ µg kg$^{-1}$, respectively. On the other hand, grain Cd concentration in CA treatment increased to $13,852 \pm 23$ µg kg$^{-1}$ compared to the control.

Fe concentration in wheat grains showed statistically significant differences depending on Cd stress and amino acid applications ($p < 0.05$). Under Cd0 conditions, the highest Fe concentration was found in Gly-treated plants (47.60 mg Fe kg$^{-1}$), while the lowest was observed in the C0 group (37.6 mg Fe kg$^{-1}$). Under Cd3 conditions, the highest Fe content was found in Asn-treated plants (55.70 mg Fe kg$^{-1}$), while the C0 group had significantly lower Fe values (39.3 mg Fe kg$^{-1}$). Under Cd12 stress conditions, the Asn-treated plants showed the highest Fe content (58.7 mg Fe kg$^{-1}$), while the lowest Fe content was recorded in the control group (46.40 mg Fe kg$^{-1}$).

Zn concentration showed similar trends to Fe in the treatments. Under Cd0 conditions, the highest Zn content was observed in the plants treated with Asn (14.5 mg Zn kg$^{-1}$), while the lowest content was recorded in the C0 group (10.40 mg Zn kg$^{-1}$). At the Cd3 dose, the highest Zn content was found in the Asn-treated plants (15.99 mg Zn kg$^{-1}$), while CA and Gly treatments showed no statistically significant difference compared to the C0 group. Under Cd12 conditions, Zn concentration was highest in the Asn treatment (55.70 mg Zn kg$^{-1}$), while the C0 group showed significantly lower Zn values (13.44 mg Zn kg$^{-1}$).

Cu concentration showed a statistically significant reduction in response to the application of amino acids (AA) ($p < 0.05$). Under Cd0 conditions, the highest Cu content was observed in the C0 group (4.20 mg Cu kg$^{-1}$), while Cu content was reduced by the application of CA (2.40 mg Cu kg$^{-1}$) and Asn (2.60 mg Cu kg$^{-1}$). At the Cd3 dose, the highest Cu concentration was observed in the C0 group (5.1 mg Cu kg$^{-1}$), while the Cu content was reduced by the application of CA (3.7 mg Cu kg$^{-1}$) and Gly (3.9 mg Cu kg$^{-1}$). Under Cd12 conditions, the highest Cu content was found in the C0 group (4.50 mg Cu kg$^{-1}$), but the application of AA and OA significantly reduced the Cu content. The lowest Cu concentration was found in the Asn treatment (2.2 mg Cu kg$^{-1}$).

Mn concentration in wheat grain also showed statistically significant variations in response to Cd stress and AA applications. Under Cd0 conditions, the highest Mn content was observed in the C0 group (37.40 mg Mn kg$^{-1}$), while Mn content decreased significantly after AA and OA applications. At Cd3 dose, the Mn content was highest in the C0 group (37.5 mg Mn kg$^{-1}$), while the lowest content was observed in CA-trated plants (24.91 mg Mn kg$^{-1}$). Similarly, under Cd12 conditions, the highest Mn concentration was observed in the C0 group (39.3 mg Mn kg-$^1$), while significantly lower Mn levels were observed in the CA (25.94 mg Mn kg$^{-1}$) and Asn (28.90 mg Mn kg$^{-1}$) treatments. These results indicate that the application of CA in particular was effective in reducing Mn accumulation in the wheat grains under Cd stress.

A pairplot (bivariate distribution plot) was generated to examine the distributional and correlational relationships among key variables under increasing Cd doses following amino acid foliar applications in wheat (Figs. 1A and 1B). This pairplot displays relationships between grain yield, N, Cd (Fig. 1A), Zn, Fe, Cu, Mn (Fig. 1B), and grain nitrogen N under different Cd doses and foliar treatments. Diagonal panels show histograms illustrating the distribution of each variable; lower left panels present scatterplots showing bivariate relationships; and upper right panels display Pearson correlation coefficients (r values) indicating the strength and direction of correlations. The analysis revealed a significant positive correlation between the Cd content in the grain and the concentrations of Zn and Fe ($p < 0.05$). Specifically, as grain Cd content increased, Zn ($r = 0.474$, $p = 0.003$) and Fe ($r = 0.498$, $p = 0.002$) levels also tended to increase along with Cd accumulation. However, no significant correlation was observed for Cu and Mn ($p > 0.05$). In addition, grain yield showed a weak positive correlation with Zn and Fe, while a weak negative correlation was found with Cu and Mn. Based on these results of the correlation analysis, the presence of high Zn, Fe, Cu or Mn concentrations does not appear to contribute to an increase in grain yield.

Foliar application of AAs and OA used in this study had a significant effect on Cd uptake and retention in wheat. Among the treatments, the foliar application of Asn was the most effective amino acid in reducing Cd accumulation in grains and resulted in the lowest Cd uptake in wheat grains. In contrast, Gly and CA showed moderate effects, with Gly having a relatively lower effect on Cd reduction and CA showing the least effect (Fig. 2).

The results of the study indicate that Cd contamination of soils has a negative effect on the yield and mineral content of wheat grains. Foliar application of AAs, especially Asn and Gly, significantly alleviated Cd-induced stress and improved grain yield. In addition, these

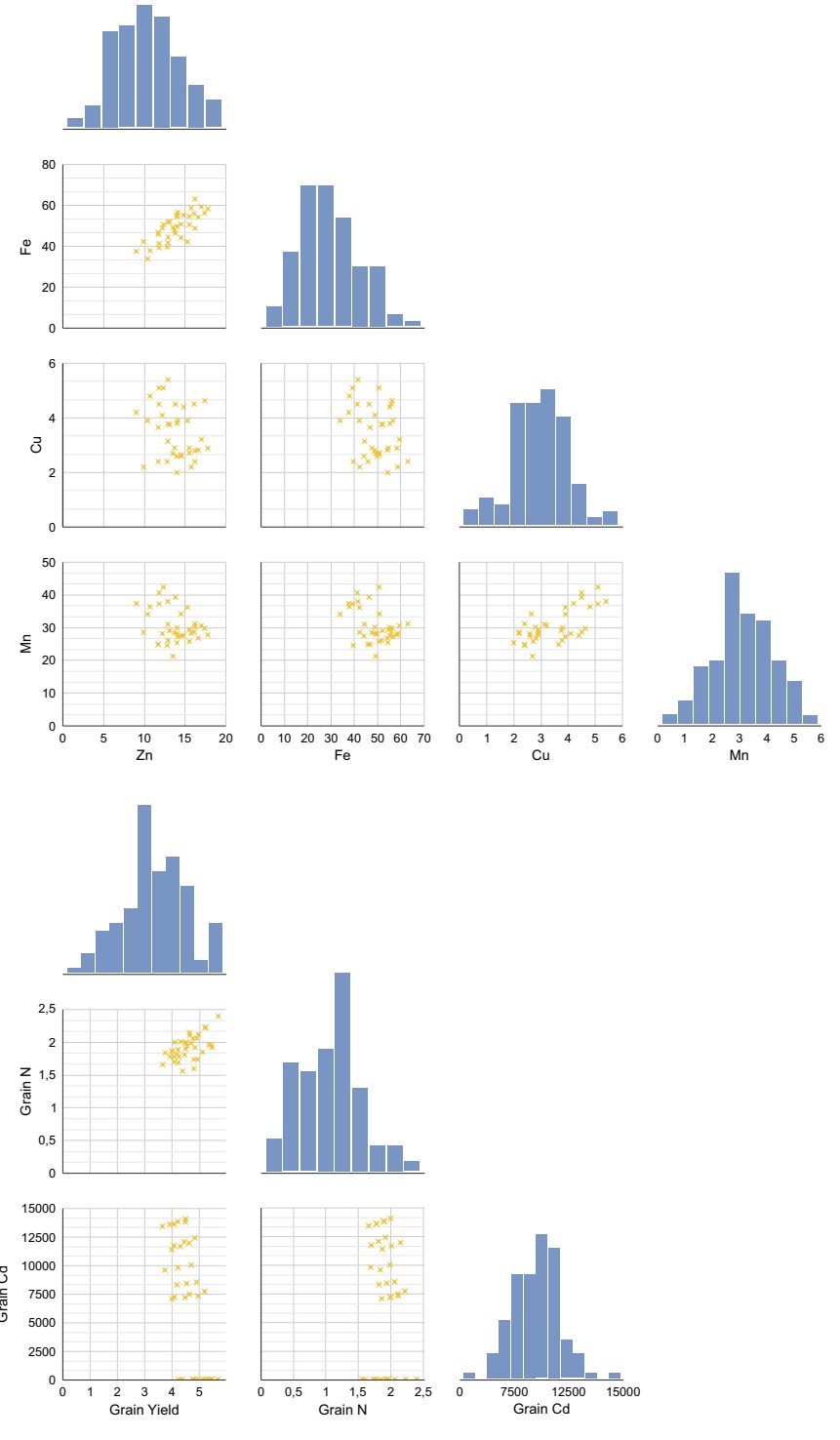

**Figure 1** The distributional relationships between variables at increasing Cd doses after AAs foliar applications in wheat.

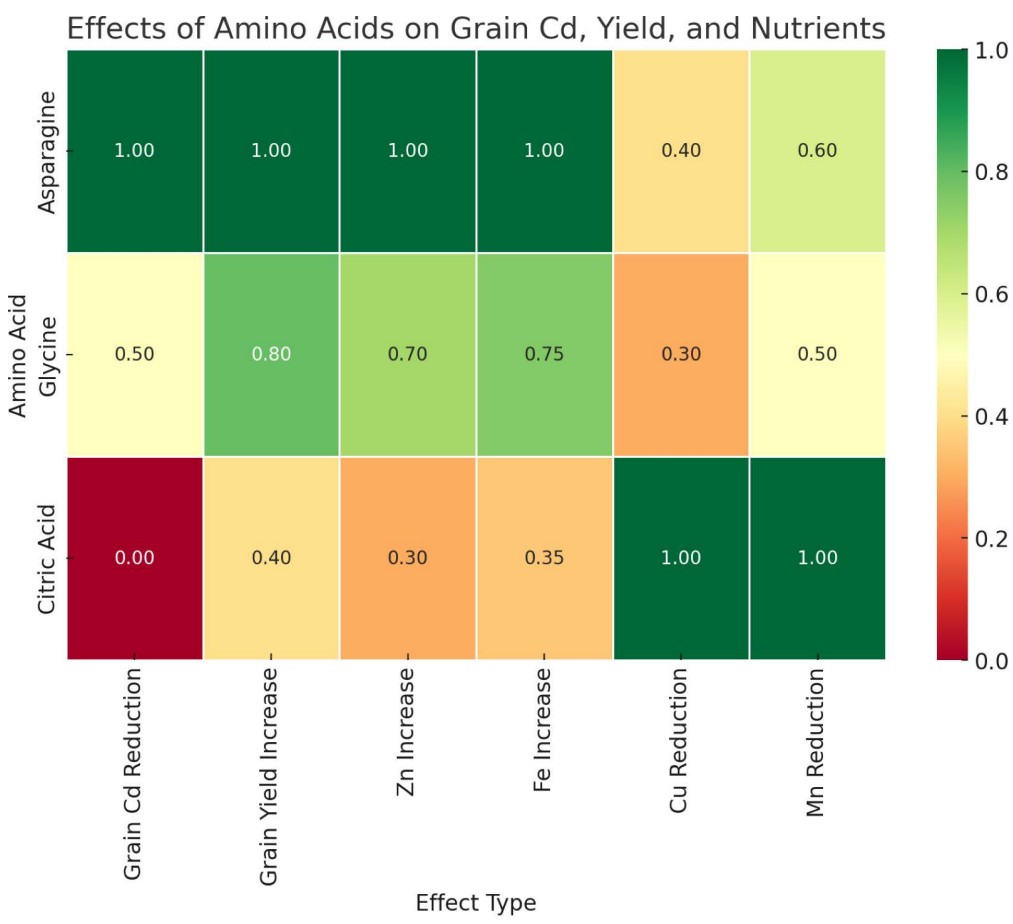

**Figure 2 Comparative effects of amino acid treatments on cadmium reduction, yield and nutrient uptake in wheat.**

treatments increased Fe and Zn concentrations, while they decreased Cu uptake under Cd stress. On the other hand, the application of CA had a positive effect on Zn uptake, but also increased Cd accumulation under Cd12 conditions.

## DISCUSSION

Results suggest that the type of amino acid plays a crucial role in reducing Cd accumulation in wheat grains. It is assumed that the effectiveness of amino acids in Cd reduction depends on their inherent properties, as also reported in the literature (*Chaffei et al., 2004*). Exogenous amino acid applications can alter Cd translocation in plants. Previous studies have shown that exogenously administered amino acids can improve metabolic functions in plants and exert a protective effect against Cd toxicity by limiting the accumulation of the heavy metal (*Badawy et al., 2024*). It was reported that the addition of Gly to a Cd-treated nutrient solution significantly reduced the Cd concentration in the xylem sap of wheat, thereby limiting Cd translocation into the shoots (*Hussain et al., 2018*; *Zhang et al., 2023*).

In contrast, certain amino acids, such as histidine, were shown to increase Cd mobilization into the shoots (*Yang et al., 2023*). This result suggests that the chemical properties of amino acids play a crucial role in determining their interactions with Cd (*Rizwan et al., 2016*). In this study, the observed reduction in Cd translocation to the grain, particularly in the Asn-treated groups, supports the hypothesis that Asn facilitates Cd retention in roots or non-edible plant parts by binding Cd or altering its intracellular distribution. This chelation mechanism could reduce the amount of free Cd available for xylem loading and subsequent transport into the grains, as previously reported (*Gill & Tuteja, 2010*; *Kocaman, 2023a*; *Aydemir et al., 2025*). The significantly lower Cd accumulation in the grains of amino acid-treated plants, compared to the control and CA groups, provides further support for this mechanism. These findings suggest that the application of specific amino acids may effectively reduce Cd mobility and accumulation in edible plant tissues.

In addition to their direct chelating effect, the application of amino acids may also have improved the internal Cd detoxification pathways in plants. Plants detoxify Cd by converting it into less sensitive tissues or chemical forms. In particular, free $Cd^{2+}$ ions are rapidly complexed in cells with thiol-rich molecules such as glutathione, phytochelatins and metallothioneins to form non-toxic Cd compounds (*Gill & Tuteja, 2010*). These Cd ligand complexes are then sequestered in vacuoles *via* specific transporters and prevent Cd from interfering with essential cellular processes (*Keunen et al., 2013*; *Zhao & Wang, 2020*; *Shi et al., 2022*). In addition, plants generally retain Cd in the roots, limiting the exposure of the xylem (*Mendoza-Cózatl et al., 2011*; *Pasricha et al., 2021*), which ultimately protects the shoots and grains from Cd accumulation (*Zhang et al., 2024*). The present study suggests that Asn, which was most effective in reducing Cd uptake, may have enhanced one or more of these detoxification steps. In addition, Asn application may have increased the ability to chelate Cd by providing glutathione precursors or triggering stress signals that enhance phytochelatin synthesis. This is supported by previous studies showing that the application of external amino acids and similar compounds can increase Cd sequestration capacity in plants and thereby mitigate Cd stress (*Farhat et al., 2022*; *Kocaman, 2023b*). For example, foliar application of certain biostimulants (*e.g.*, ascorbic acid) in wheat has been reported to reduce Cd levels in shoots by more than 50% (*Farhat et al., 2022*). This suggests that exogenous supplementation could enhance the plant's internal Cd detoxification mechanisms and the plant's ability to fix Cd in the roots or in immobilized form. Overall, results support the idea that the administered amino acids act either as Cd chelating agents or as metabolic enhancers that minimize Cd uptake and promote sequestration into safer compartments. Thus, the application of AA and OA proves to be an effective strategy to reduce Cd accumulation in wheat grains and contributes to safer agricultural production in Cd-contaminated soils.

The results of the study show that foliar application of AAs and OAs not only reduced Cd accumulation, but also significantly improved the growth and grain yield of wheat under Cd stress conditions. The treated plants produced higher grain yields compared to the C0 groups. In particular, the application of Asn resulted in the greatest yield increase, suggesting that it plays a crucial role in stress mitigation and growth promotion. AAs are known to serve as readily available N sources and metabolic building blocks for plants.

Plants that take up exogenously supplied amino acids save the energy they would otherwise have to spend on synthesizing these molecules and can redirect this energy into growth and grain filling. In other words, amino acid supplementation can reduce the metabolic burden of N assimilation, increase the efficiency of nutrient utilization and improve the development of yield components. Previous studies have reported that foliar application of amino acids increases crop yield by optimizing nutrient uptake and utilization (*Popko et al., 2018*). Similarly, a field study with wheat showed that amino acid–based biostimulants increased grain yield by 5% to 11%, which was attributed to improved nutrient supply and plant vigor (*Popko et al., 2018*).

Foliar applications had a significant effect on the uptake and accumulation of essential micronutrients in wheat grains, especially on the concentrations of Zn, Fe, Cu, and Mn. Significant changes were observed, especially for these micronutrients. In general, foliar treatments resulted in higher N, Zn and Fe concentrations in the grains compared to the C0 groups, while Cu and Mn concentrations showed minor fluctuations or constant increases. These pronounced changes in the nutrient composition of wheat grains indicate that the reduction in Cd accumulation improves the plant's ability to take up micronutrients and translocate them efficiently into the grains. Cd in soil can inhibit the uptake of nutrients such as Zn and Fe because Cd competes with these elements for the same transport pathways, leading to nutrient deficiency le. It is known that $Cd^{2+}$ ions enter plant roots *via* transporters that normally facilitate the uptake of $Zn^{2+}$, $Fe^{2+}$, $Cu^{2+}$ and $Mn^{2+}$ (*Abedi & Mojiri, 2020*). If the Cd content in the soil is too high, these transport channels can therefore become saturated or blocked, limiting the uptake of important nutrient ions (*Tarakçıoğlu, Aşkın & Kızılkaya, 2006*). In addition, excessive Cd accumulation can trigger physiological responses that mimic Fe or Zn deficiency, leading to symptoms such as chlorosis and the overexpression of Fe transport genes (*Schaaf et al., 2006*; *Morrissey et al., 2009*; *Fukao et al., 2011*; *Wu et al., 2012*). Due to these competing interactions and deregulatory mechanisms, high Cd concentrations are often associated with lower micronutrient concentrations in plant tissue.

The results of the study suggest that exogenous foliar application likely attenuates competitive inhibition through reduced Cd uptake, allowing for better nutrient uptake and turnover. In addition, the wheat grains of foliar-treated plants exhibited lower Cd accumulation, allowing the uptake systems for Zn, Fe, Cu, and Mn to function more normally. The observed increase in Zn and Fe concentrations in the Asn-treated groups further supports this result. Previous studies have reported an inverse relationship between Cd and Zn, suggesting that increased Zn availability in plants often leads to decreased Cd accumulation (*Sarwar et al., 2015*). Conversely, a reduction in Cd content in plants can improve Zn uptake. The same study also showed that foliar fertilization with Zn reduced the Cd content in wheat grains by 36%, while the concentrations of Zn, Fe, and Mn were significantly increased (*Lu et al., 2023*). In *Lu et al. (2023)*, exogenous foliar fertilization reduced the Cd content in the grains, while it attenuated the Cd-induced inhibition of Zn and Fe translocation into the grains. This result is particularly important for nutritional quality, as higher Zn and Fe concentrations improve the nutritional value of wheat grains. Results suggest that foliar applications may contribute to this process by

modulating Cd–nutrient interactions. In addition, exogenously applied Gly betaine has been shown to mitigate oxidative damage in Cd-stressed plants and improve plant height and root length (*He et al., 2019*). Similarly, foliar spraying of various plants with amino acids improved growth and yield indices by mitigating the effects of abiotic stress (*Wang et al., 2019*). Overall, exogenous foliar applications effectively reduced Cd toxicity. The observed synergy between lower Cd levels and increased micronutrient concentrations not only minimizes Cd-related health risks, but also improves the nutritional quality and bioavailability of essential minerals in wheat grains, which represents a significant agronomic and nutritional benefit (*Lu et al., 2023*; *Özkutlu et al., 2025*).

In summary, the results of the study are consistent with the literature, which indicates that the application of amino acids has a biostimulant effect and can increase yield under both optimal and stress conditions (*Popko et al., 2018*; *Shahrajabian, Cheng & Sun, 2022*). By facilitating nutrient uptake and mitigating Cd toxicity, wheat plants were able to maintain growth and fill their grains more efficiently, which explains the observed yield increases. Similarly, previous studies have reported that exogenously administered proline and Gly betaine alleviated Cd stress, reduced Cd accumulation and improved plant growth in wheat and other crops (*He et al., 2019*). This study focused specifically on Asn and Gly. Asn is known to accumulate in plants under heavy metal stress (*Zhao et al., 2006*), probably as part of a detoxification or N storage response. *Chaffei et al. (2004)* reported that tomato plants exposed to Cd stress redirected N into free amino acids (particularly glutamine/Asn) rather than proteins as a protective strategy (*Chaffei et al., 2004*). In this context, the application of Asn in study may have reinforced a natural tolerance mechanism. Although direct prior data on Asn applications are limited, results suggest that this amino acid could be incorporated into broader Cd stress mitigation strategies alongside other amino acids, offering a potential way for enhancing plant resilience in Cd-contaminated environments.

Another compound examined in the study by *Özkutlu et al. (2025)*, CA, offers an interesting comparison with previous studies. Unlike amino acids, CA is an organic acid with strong metal chelating properties, and its role in Cd dynamics is somewhat complex. Some studies have reported that the addition of CA to soil increases the bioavailability of Cd and thus promotes Cd accumulation in plants (*Li & Zhou, 2012*). In another study, the application of CA in the root zone was found to dissolve Cd in the soil and exacerbate Cd toxicity in wheat (*Huo et al., 2016*). Conversely, other studies suggest that the application of CA to the leaves can reduce the transfer of Cd to the grains (*Xue et al., 2021*; *Islam et al., 2022*; *Chen et al., 2024*). For example, spraying rice leaves with CA was reported to enhance Cd sequestration and suppress Cd translocation by increasing the competitive interaction with Mn (*Chen et al., 2024*). The CA results in *Özkutlu et al. (2025)* are more consistent with this second scenario. Since CA was applied through the leaves and not directly to the soil, it may have immobilized Cd in the plant instead of mobilizing it in the soil. The experiment showed that the application of CA reduced the Cd content in the wheat grains, suggesting that it could be a means of reducing Cd if applied correctly.

While soil-applied CA can release Cd ions so that they are more available for uptake by the roots, foliar-applied CA can interact with the Cd already present in the plant and convert it into less mobile or less toxic forms (*Lakkakula, Lima & Walker, 2004*). Overall,

the study contributes new data to the growing literature on bio-organic solutions to heavy metal stress. The results support the potential of amino acid–based interventions as effective tools to manage crop production in Cd-contaminated soils and represent a promising approach to mitigate heavy metal toxicity in plants.

## CONCLUSION

The comparative evaluation of Asn, Gly and CA in this study identified Asn as the most effective treatment for improving the performance of wheat under Cd-contaminated conditions. This AA significantly reduced Cd accumulation in wheat grains, thus minimizing the risk of contamination of the food chain. It also increased grain yield and most strongly promoted plant growth and productivity under stress conditions. In addition, Asn had a positive effect on micronutrient profiles in the grains by preserving or increasing essential nutrients such as Zn and Fe while suppressing Cd accumulation. Considering its role in heavy metal reduction, yield increase and nutrient quality improvement, Asn proves to be a promising biostimulant for wheat cultivation in Cd-contaminated soils.

From a practical point of view, foliar application of Asn during critical growth stages could be an effective agricultural strategy to ensure crop safety in fields with light or moderate Cd contamination. Farmers and agricultural practitioners could integrate this amino acid treatment into their regular fertilization and phytoremediation programs to develop a more sustainable management strategy for contaminated areas. However, optimization of the application method and dosage is crucial. The concentration and timing of application used in this study have been shown to be effective. However, transferring this protocol to field conditions requires on-farm trials to ensure maximum benefit and cost-effectiveness.

Further research is recommended to expand on these results. Future studies could investigate the biochemical and molecular mechanisms underlying Asn-mediated Cd tolerance, for example, by analyzing gene expression of Cd transporters and ligands or changes in root exudate. In addition, it would be beneficial to investigate the combination of AAs with other ameliorants such as organic matter or mineral nutrients such as Zn, as integrated approaches could have synergistic effects on Cd reduction and yield enhancement. Furthermore, extension of this research to different wheat varieties or other agricultural crops is essential to determine the general applicability of amino acid treatments to mitigate heavy metal stress.

In conclusion, this study represents a promising and environmentally friendly strategy to improve crop safety and productivity in Cd-contaminated soils. Asn proves to be a valuable tool in the effort to produce cleaner, nutrient-enriched wheat grains and strengthens its potential as a practical bio stimulant for sustainable agriculture.

### Funding

The author received no funding for this work.

## Competing Interests

The author declares there are no competing interests.

## Author Contributions

- Özlem Ete Aydemir conceived and designed the experiments, performed the experiments, analyzed the data, prepared figures and/or tables, authored or reviewed drafts of the article, and approved the final draft.

## Data Availability

The raw data is available in the Supplemental Files.

## Supplemental Information

Supplemental information for this article can be found online at http://dx.doi.org/10.7717/peerj.20102#supplemental-information.

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
