# Peer review of "Effects of foliar applied asparagine, glycine and citric acid on cadmium uptake and yield in wheat"

_PeerJ, doi:10.7717/peerj.20102_

## Round 0.1 · original submission · Major Revisions

· Academic Editor

Major Revisions

The review process for your manuscript has been completed. Please revise your manuscript taking into account reviewer’s comments as well as my recommendations below. Please note that failure to make and/or explain corrections may result in your manuscript being rejected.

The title is too long. Replace it with a more effective and shorter title.
Abstract Line 31: An ambiguous sentence. Is cadmium pollution a heavy metal? Or is cadmium a heavy metal that causes pollution that threatens agricultural production and food safety?
The abstract should briefly mention how asparagine, glycine, and citric acid applications are performed. The results are presented in a disjointed manner. A sense of coherence should be established.
Why are there no spaces when references are given in parentheses in the Introduction section? This is incorrect. The reference format is completely wrong. Özkutlu should not be the only person working on Cd pollution in the world, and these citations are probably things he got from someone else. Please cite them appropriately.
Line 53: Abbreviations should be explained the first time they appear.
Lines 72-85: Since you mention bread wheat (T. aestivum), you probably used it in your study. Why did you feel the need to compare it with durum wheat? If there is no such comparison in the study and if you have no findings related to oxidative stress, it is unnecessary to provide this information in the introduction. Instead, write the effect of Cd on the parameters you measured in your study and the effect of your applications on reducing Cd stress.
Line 131: Where does this parenthesis close?
Line 133: What does “The experiment was conducted randomly” mean? Please specify a scientifically valid experimental design. And how many replication? Farmer's field???
Line 136: If durum wheat (Triticum durum) is the experimental material, why is there a sentence in the introduction that says, “Wheat (Triticum aestivum L.) is one of the world's most widely cultivated staple foods and the...”? Please refer to durum wheat and explain why you chose the Harran 95 variety.
Line 138: What were the exact dimensions of the plastic pots? A small area and insufficient plants to measure grain yield.
Line 144: “Field capacity” is true “field strength.” is not. “irrigated.” is true. ‘Watered’ is not.
Why are there three different Cd doses?
Applying every five days is not practical.
Is it sufficient to only analyze at N among the macroelements? The others????
Lines 244-258: Are you sure this section should be included in the Discussion section? There are no explanations referring to Figures 1 and 2 in the Results section.
Line 269: What does the number 8 above the Cd symbol represent?
Line 272: It would be more appropriate to have data to support this statement.
Line 304: biomass?? I did not see any data related to this.
Line 326: Ca or Cu?
Line 357: Which literature? Please add them.
The discussion section seems more like a sharing of textbook information than a discussion of your research findings. Even if you try to base it on the results, there is no logical pattern. Is the increase in microelements in the seeds sufficient? Or is Cd decreasing to an acceptable level? Explain these based on the literature. Avoid unnecessary repetition. Remember that each amino acid may behave differently, and focus on your own applications when discussing the results. Additionally, emphasize the effects of these amino acids and why you chose them in the introduction section rather than in general terms.
Are you sure that Figure 1 shows the relationships between the characteristics?

**Language Note:** The review process has identified that the English language must be improved. PeerJ can provide language editing services - please contact us at [email protected] for pricing (be sure to provide your manuscript number and title). Alternatively, you should make your own arrangements to improve the language quality and provide details in your response letter. – PeerJ Staff

·

Basic reporting

1. Where is the text corresponding to figure 1? Explain the components/features/parameters shown in figure 1 in detail in the corresponding legend. For example, what Grain N denote for? Cite this figure in the text. Explain what is the analysis the graph is showing. It appears like correlation analysis, but why are there bra graph through the diagonal. What do X- and Y- axes show? What do the x marks and bars show?
2. What do a, b and c letters denote in table 2? The Author mentioned that these letter indicate significant p value. But what is the difference between three letters?
3. How are the values shown in figure to computed? Was the data normalized to 'no Cd' or 'no treatment'?
4. Authors did not explain how the randomization of treatment groups happened in detail. Please elaborate it in methods section. What factors were considered as potential confounders and how were they taken care of in your randomized design?
5. Did you measure levels of cadmium in each pot at the end of the experiment?

Experimental design

1. The author did not report any of the observation that could possibly explain the reason behind the decrease in yield after Cd exposure and the restoration of yield following leaf application. Author should catalogue the following features that may explain the observed differences in the yield.
a) Differences in time of onset and density of flowering in each group
b) Differences in number of spikelets and their average length in each group
c) Fertility frequency in apical, central and basal portions of spikelets.

2. What is the rationale for selecting the Cd concentrations used in the study? This should be explained. The concentration levels used in the experiments should also be compared to contamination levels seen agricultural settings.

3. Please explain number plants in each group in the experiment. Please explain how the yield was measured? Did you count the number grains per plant?

Validity of the findings

1. Please provide methodology of ICP-OES and how cross contamination was avoided during the experiment.

Reviewer 2 ·

Basic reporting

The expression is written in an understandable way.
The number of tables and graphs is quite low. The figures are not clear.
The number of literature is sufficient, but the expression would be enriched if it were differentiated.
Aspargine, glycine and citric acid application dose and time can be added in the summary section.
In in-text references, add a space between the previous word and the source.
In all references starting with Özkutlu, they should all be written the same way in order to ensure uniformity. (Not as Özkutlu or Ozkutlu, just Özkutlu)
In sequential references in the text, sources should be given in year order.
In sequential expressions, put a comma before "and". (for example, A,B, and C)

Experimental design

Methods described with sufficient detail & information to replicate.
It is not clear from which table or figure the data written in the results were taken. should be added.

Validity of the findings

The discussion and conclusion sections are too long and there are too many similarities, they should be shortened and corrected.

Additional comments

Dear Author, I would like to ask you to make the corrections in the attached file and re-read the text you wrote. The language used is quite understandable, but the repetitive expressions make it difficult to read and understand the text. Also, if you had looked at more agronomical characters, the content of your study could have been enriched.

Annotated reviews are not available for download in order to protect the identity of reviewers who chose to remain anonymous.

Reviewer 3 ·

Basic reporting

The article is well written in English. But some references are given incorrectly. For example, Line 50, which is the first sentence of the Introduction section, cites the reference (Kocaman 2023b). Line 58, Özkutlu and Kara (2019) repeated twice. A similar mistake has been found in Line 69.
In the introduction section, Line 52, to clarify the Cd contamination, you should specify the area with Cd-contaminated soils. The readers need to have knowledge about the contamination of Cd in agricultural areas.

Experimental design

The experimental design is absent. It must be inserted into the M&M section.
Line 131. No need to coordinate the greenhouse, but it is necessary to state where the experiment was conducted, including the country, city, etc.
Line 133. "The experiment was conducted randomly." What does it mean?
Line 148. What was the control treatment for foliar spraying?
Line 152. Foliar applications were performed five times at a 5-day interval. This is a practical use for farmers or producers.
Line 136. Please explain why you prefer the Harran95 variety. Also, give information about its vernalization requirements.
Please give a detailed explanation about the irrigation regimes.
Line 169. What was the experimental design?
Figure 1 is very complex and incomprehensible.

Validity of the findings

weak.

---

## Round 0.2 · Minor Revisions

· Academic Editor

Minor Revisions

Thank you for your corrections. Your manuscript looks quite good as is. However, you should also make the following minor revisions before acceptance.

Line 33: Italicize Triticum durum.

Lines 34-35: Write the abbreviations glycine, asparagine, and citric acid. You already abbreviated them in lines 31-32. Do this for the entire abstract. If you want to write a long form, don't use any abbreviations at all.

Line 55: Delete Cr (Chromium), Hg (Mercury), Pb (Lead). Simply write Cd and the others.

Line 56: Write only the abbreviation Cd. You already abbreviated it in the previous line. Pay attention to these abbreviations throughout the text and correct them. Write the abbreviation in parentheses only at the first occurrence, and then use only the abbreviation. For example, in lines 61 and 63, you wrote cadmium in long form again. Carefully review the entire text and correct any abbreviations you made.

Line 84: Only the short form Cd.

Line 114: amino acid and organic acid. If you want to use abbreviations, use abbreviations from here onwards throughout the text. Or don't use any abbreviations at all.

Line 140: Write "N and microelement uptake" here. You also examined N, and it's not a microelement. In your response letter, include the following sentences in the introduction, along with references, explaining why you measured N.

Line 219: You've mentioned asparagine (Asn), glycine (Gly), and citric acid (CA) many times before. But you didn't abbreviate them. Either abbreviate them at the first mention, then only use the abbreviation, or always write the long form.
Write something about Figures 1 and 2 in the Results section. It's not appropriate to mention them only in the Discussion section. You will discuss your findings related to them in the Discussion section.

Line 292: Write "N," not "nitrogen."

Lines 347, 421, 430, and 432: Do not write "our results." Write the results of this study. You are the sole author.

·

Basic reporting

Authors addressed my concerns.

Experimental design

.

Validity of the findings

.

Reviewer 2 ·

Basic reporting

In the citations within the text, write the sequential ones in a single parentheses and list them in order of year from largest to smallest.
Please pay attention to the italicization of Latin names in the references section.
The line spacing of some paragraphs within the text does not comply with the general text. This should be checked.

Experimental design

Research question well defined, relevant & meaningful. It is stated how research fills an identified knowledge gap

Validity of the findings

Conclusions are well stated, linked to original research question & limited to supporting results

Additional comments

Thank you for considering the corrections I previously mentioned. Please re-examine the typos and minor corrections, especially in the in-text references.

---

## Round 0.3 · accepted · Accept

· Academic Editor

Accept

Your manuscript is accepted after the last corrections.